# The Tumor Microenvironment: Signal Transduction

**DOI:** 10.3390/biom14040438

**Published:** 2024-04-04

**Authors:** Xianhong Zhang, Haijun Ma, Yue Gao, Yabing Liang, Yitian Du, Shuailin Hao, Ting Ni

**Affiliations:** 1State Key Laboratory of Reproductive Regulation and Breeding of Grassland Livestock, Institutes of Biomedical Sciences, School of Life Sciences, Inner Mongolia University, Hohhot 010070, China; zhangxianhong96@163.com (X.Z.); gyueyue1231@163.com (Y.G.); liangyabing9099@163.com (Y.L.); 111996007@imu.edu.cn (Y.D.); 2Key Laboratory of Ministry of Education for Protection and Utilization of Special Biological Resources in Western China, School of Life Sciences, Ningxia University, Yinchuan 750021, China; mahj@nxu.edu.cn

**Keywords:** tumor microenvironment, signaling pathways, tumor metabolism, tumor immunity

## Abstract

In the challenging tumor microenvironment (TME), tumors coexist with diverse stromal cell types. During tumor progression and metastasis, a reciprocal interaction occurs between cancer cells and their environment. These interactions involve ongoing and evolving paracrine and proximal signaling. Intrinsic signal transduction in tumors drives processes such as malignant transformation, epithelial-mesenchymal transition, immune evasion, and tumor cell metastasis. In addition, cancer cells embedded in the tumor microenvironment undergo metabolic reprogramming. Their metabolites, serving as signaling molecules, engage in metabolic communication with diverse matrix components. These metabolites act as direct regulators of carcinogenic pathways, thereby activating signaling cascades that contribute to cancer progression. Hence, gaining insights into the intrinsic signal transduction of tumors and the signaling communication between tumor cells and various matrix components within the tumor microenvironment may reveal novel therapeutic targets. In this review, we initially examine the development of the tumor microenvironment. Subsequently, we delineate the oncogenic signaling pathways within tumor cells and elucidate the reciprocal communication between these pathways and the tumor microenvironment. Finally, we give an overview of the effect of signal transduction within the tumor microenvironment on tumor metabolism and tumor immunity.

## 1. Introduction

As normal cells transform into tumor cells or tissues, they acquire a set of distinctive capabilities. These markers encompass the capability to sustain signaling for proliferation, evade inhibitors of growth, resist programmed cell death, achieve replicative immortality, activate invasion and metastasis, reprogram cellular metabolism, and evade immune-mediated destruction [1]. The most fundamental feature of cancer is its ability to sustain continuous proliferation. Abnormal expression of carcinogenic mutations or components in the signal transduction pathways within tumor cells, in contrast to normal cells, disrupts the regulatory networks that govern cellular functions. This disruption allows tumor cells to dysregulate mitotic control, resist apoptosis, and invade neighboring tissues, thereby ensuring their survival. These signaling pathways include the regulation of cell growth, division cycle, maintenance of cell metabolic homeostasis, and monitoring of immunity. Once tumor cells acquire these distinctive marker abilities, they transmit these signals to the surrounding heterogeneous cells, thereby influencing the biological characteristics and functions of other cells [2]. It is worth noting that during the progression and metastasis of tumors, the energy needed for abnormal proliferation is obtained through alterations in metabolic pathways. The molecules generated and released due to the metabolic reprogramming of tumor cells contribute to changes in cytokines or components within the tumor microenvironment. Metabolites released by tumor cells serve as signaling molecules, activating intrinsic carcinogenic signaling pathways within cancer cells. These metabolites orchestrate alterations in critical signaling pathways, influencing cancer progression by regulating processes such as proliferation, motility, survival, and angiogenesis [3]. In addition, tumor metabolites in the tumor microenvironment mediate genetic and epigenetic changes that enable tumor cells to evade tissue-based proliferation control by maintaining major pro-survival and pro-proliferation signals in a continuous “on” state [4].

Proliferating cancer cells exhibit abnormal coordination with the tumor microenvironment (TME) through recruitment and reprogramming of non-cancerous host cells, as well as restructuring of the vasculature and extracellular matrix (ECM) [5]. The acquisition and maintenance of cancer hallmarks rely on contributions from the TME, encompassing aspects such as sustaining proliferative signaling, evading cell death, inducing angiogenesis, activating invasion and metastasis, instigating pro-tumor inflammation, and avoiding immune destruction [6]. The TME can promote intercellular communication by releasing paracrine signals from cytokines, chemokines, growth factors, and proteases. In addition to direct signal communication between tumor cells, cellular molecules secreted in response to cell stress in the tumor cells’ cancer-promoting signaling pathways can exert direct and indirect effects on target cells. This occurs through binding to their receptors or remodeling the extracellular matrix [7]. Oncogenic transformation enables cancer cells to adopt a well-defined metabolic phenotype, known as metabolic reprogramming. This altered metabolic state significantly influences the TME [8]. Interestingly, the aberrant metabolism of tumor cells not only generates an anoxic or acidic environment within the TME but also modulates signaling pathways in the tumor microenvironment through metabolites. The TME is intricately connected with numerous signaling pathways in tumor cells, forming a complex network. Various external and internal signals activate and integrate signaling pathways, leading to the execution of diverse cellular functions, such as cell growth, motility, cell structure and polarity, differentiation, programmed cell death, protein synthesis, etc. [9]. Furthermore, the primary components of the TME, immune cells, stromal cells, and ECM are in close contact with cancer cells. Numerous substances released by immunological and tumor cells are responsible for immunosuppression, persistent inflammation, and the angiogenesis-promoting tumor microenvironment. In such an environment, cancer cells possess the ability to evade host immune surveillance and immune cell attacks. They gradually reshape and adapt to the microenvironment by regulating various signaling pathways [10,11]. Therefore, for efficient tumor therapy, it is crucial to understand the oncogenic signaling pathways of tumor cells as well as the signal crosstalk mediated by different cytokines in the TME.

## 2. Dysregulated Signaling Pathways in Tumor Cells

Mutations or expression changes in specific genes give tumor cells the ability to divide and grow abnormally. The acquisition of these abilities is closely related to the transmission of sustained proliferation signals (Figure 1). Alterations in genes or pathways controlling cellular fate can disrupt the cell signaling system, bestowing malignant behavior on tumor cells. Even if the mutation-activated pathway is partially blocked, tumor cells may escape growth inhibition by activating other pathways [9]. Many signaling pathways within tumor cells interact to generate complex networks. Tumor cells integrate multiple signals when stimulated by internal and external stressors and maintain rapid proliferation and development through constant activation and evolution of signaling pathways.

### 2.1. Epidermal Growth Factor Receptor (ErbB/EGFR) Signaling Pathway

The human epidermal growth factor receptor (HER/ErbB) family is composed of four members: HER-1 (EGFR/ErbB1), HER-2 (ErbB2), HER-3 (ErbB3), and HER-4 (ErbB4). Receptor tyrosine kinases belonging to the ErbB family control a wide range of signal transduction and initiate a large number of intracellular pathways [12]. Epidermal growth factor (EGF) binding on the cell surface causes the EGFR to dimerize, which activates EGFR tyrosine kinase activity and causes the receptor to trans-autophosphorylate [13]. Large signal transduction complexes are formed when downstream signal transduction proteins engage with tyrosine autophosphorylation sites in the active EGFR. The receptor signaling protein complex then initiates the activation of various signaling pathways, which promote cell growth and survival. For example, ligand EGF, heparin-binding EGF (HB-EGF), transforming growth factor α (TGF-α), bidirectional regulatory proteins, epigenes, betacellulin (BTC), and epiregulatory proteins (EPR) bind to EGFRs, while the binding of neuregulin (NRG-1, NRG-2, NRG-3, and NRG-4) to ErbB3 and ErbB4 receptors (except ErbB2) leads to homologous/heterodimerization, the reverse phosphorylation of tyrosine residues in cytoplasmic domains, and the subsequent activation of several downstream pathways. This involves the signaling pathways for phosphatidylinositol 3-kinase (PI3K)/protein kinase B (AKT), signal transducer and activator of transcription (STAT), mitogen-activated protein kinase (MAPK), extracellular signal-regulated kinase (ERK), phospholipase C-γ (PLC-γ), and RAS [14]. It is important to remember that EGFR overexpression and mutations have been linked to poor prognosis, treatment resistance, tumor survival, and metastasis in a variety of cancer types [15,16,17].

### 2.2. PI3K/AKT/Mechanistic Target of Rapamycin (mTOR) Signaling Pathway

The PI3K/AKT/mTOR signaling system is linked to other signaling pathways to facilitate tumor growth and metastasis. These include the phosphorylation of intracellular enzyme activity domains and signal transduction junction proteins, which activate PI3K, and the binding of PI3K to growth factor receptors (such as EGFRs). This activation catalyzes the phosphorylation of the 3-hydroxyl group of phosphatidylinositol 4,5-biphosphate (PIP2), generating phosphatidylinositol 3,4,5-trisphosphate (PIP3). Acting as the second messenger, PIP3 recruits 3-phosphoinositide-dependent kinase 1 (PDK1) and AKT proteins to the plasma membrane, causing PDK1 to phosphorylate threonine at position 308 of the AKT proteins (T308). This process leads to structural changes and the activation of AKT proteins. Phosphorylation then activates or inhibits several downstream substrates, regulating cell survival, growth, differentiation, metabolism, and cytoskeletal recombination in response to a variety of signals, including growth factor receptor tyrosine kinase (RTK) and G- protein-coupled receptor signaling [18]. Notably, once activated, AKT phosphorylates a large number of targets in the cytoplasm and nucleus and triggers several downstream substrates that control cell division, ultimately controlling the development and multiplication of tumor cells. For example, AKT increases tumor cell survival and prevents apoptosis by inhibiting BAD and BAX, members of the pro-apoptotic gene Bcl-2 family, and by negatively regulating FOXO and other forkhead transcription factors [19]. In addition, the AKT phosphorylation of tuberous sclerosis complex subunit 2 (TSC2) blocks the binding of TSC2 and tuberous sclerosis complex subunit 1 (TSC1), causing the liberation of an RAS homolog protein enriched in brain (RHEB) from inhibitory control. This release indirectly activates the mechanistic target of rapamycin complex 1 (mTORC1) through RHEB. Once activated, mTORC1 phosphorylates its effector 70kDa ribosomal protein S6 kinase (p70S6K) and eukaryotic translation initiation factor 4E-binding protein 1 (4E-BP1), thereby initiating processes such as transcription, proliferation, growth, and protein synthesis in tumor cells [20].

The activation of the PI3K/AKT/mTOR signaling pathway is prevalent in tumorigenesis, predominantly attributed to the dysregulation of various components of the signaling pathway at different levels, encompassing DNA (including gene amplification, deletion, mutation, and fusion), RNA (transcription and post-transcriptional regulation), and protein (including protein stability control and post-translational modification) [21]. Recent studies have highlighted a high frequency of mutations in genes such as phosphatidylinositol-4,5-bisphosphate 3-kinase, catalytic subunit alpha (PIK3CA), phosphoinositide-3-kinase regulatory subunit 1 (PIK3R1), phosphatase and tensin homolog (PTEN), and AKT within the PI3K/AKT pathway, establishing significant associations with tumor initiation, progression, and drug resistance [22]. Moreover, the PIK3CA mutation can lead to prostate cancer in mice and is associated with a poor prognosis [23,24,25]. In addition, PTEN is well-demonstrated as a regulator of tumor cell proliferation and survival, primarily by instigating cell cycle arrest, predominantly through its cytoplasmic impact on the PI3K-AKT pathway [26]. PTEN functions by dephosphorylating the second messenger PIP3 in tumor cells, converting it back to PIP2, and thereby interrupting the PI3K signaling pathway. Therefore, PTEN is often considered to be a tumor suppressor. Instances of PTEN expression deletion in tumor cells are commonly observed, often involving insertions, which alter the reading frame and promote premature termination, deletion, or promoter methylation. Many of these mutations have been found in many types of tumors, especially metastatic human cancers [27]. The aberrant activation of PI3K/AKT signaling results from gain-of-function mutations in PI3K/AKT and loss-of-function mutations in the tumor suppressor PTEN. In addition, mutations in the *mTOR* gene itself, which is a downstream target of PI3K/AKT, sustain the mTOR signaling pathway in a hyper-activated state, and regulate the growth and metabolism of tumor cells [28]. Moreover, alterations in the constituent genes of the two complexes constituting mTOR, mTORC1, and mechanistic target of rapamycin complex 2 (mTORC2), also contribute to the activation of the PI3K/AKT/mTOR signaling pathway, influencing the growth of tumor cells. It is worth noting that mTOR signaling is usually activated in tumors to control cancer cell metabolism by altering the expression or activity of many key metabolic enzymes [29].

### 2.3. RAF/Mitogen-Activated Extracellular Signal-Regulated Kinase (MEK)/ERK and MAPK Signal Pathways

The ErbB/EGFR signaling pathway activates not only the PI3K/AKT pathway but also the RAF/MEK/ERK and MAPK signaling pathways [30]. The initiation of the MAPK signaling cascade occurs when RTK and RAS are activated. Like MAPK, MKK is also negatively regulated by phosphatase, so MKK is also an upstream regulatory kinase. RAF kinase belongs to the MAPK kinase kinase (MKKK) family and can act as a direct regulator of MKK. Signal transduction in the RAF-MEK-ERK pathway is initiated by binding multiple ligands to RTK, especially growth factor receptors such as EGFR. Gtp-loaded RAS recruits and activates RAF, leading to recruitment of RAF to the plasma membrane, dimerization, and subsequent phosphorylation. After the phosphorylation of RAF, the bispecific protein kinase MEK1/2 is activated, which further leads to the phosphorylation of T202/Y204 sites of ERK1 and T183/Y185 sites of ERK2, thus altering the expression of downstream target genes. Mutations in components of the RAS-RAF-MEK-ERK signaling pathway have been linked to different types of cancer [31,32,33,34,35]. Among them, RAS, encoded by three universally expressed genes (*RAS*, *KRAS*, and *NRAS*), serves as a crucial protein governing cell proliferation, differentiation, and survival [36]. Distinct RAS mutants are associated with differences in patient survival. For instance, Kirsten rat sarcoma viral oncogene homologue (KRAS) is frequently mutated in pancreatic, colorectal, lung, and biliary cancers, while Harvey rat sarcoma viral oncogene homolog (HRAS) and neuroblastoma RAS viral oncogene homolog (NRAS) exhibit a higher frequency of mutations in salivary gland tumors and melanoma [37]. In addition, B-Raf proto-oncogene (BRAF) is frequently mutated in melanoma, thyroid cancer, colorectal cancer, and ovarian cancer [38]. Therefore, the RAF-MEK-ERK signaling pathway plays an important role in tumorigenesis, maintenance, and metastasis. Key components of this pathway, namely RAF, MEK, and ERK, emerge as appealing targets for the development of potential anti-cancer therapies.

### 2.4. p53 Signaling Pathway

The tumor suppressor p53, encoded by *TP53*, plays a crucial role in normal cell growth and tumor prevention [39]. The p53 protein exhibits a complex structure, including a trans-activation domain (TAD), proline-rich region (PRR), central DNA-binding domain (DBD), tetramerization domain (TD), and carboxyl-terminal domain (CTD). Functioning as a transcription factor, p53 actively engages in numerous essential biological processes by regulating the expression of downstream target genes, such as cell cycle arrest, cell apoptosis, and a series of anti-proliferation processes. Notably, in response to stressful conditions or oncogene activation, elevated p53 levels regulate apoptosis in tumor cells by regulating the p53 upregulated modulator of apoptosis (Puma) and Noxa [40,41]. Similarly, p53 can also regulate the cell cycle by regulating the expression of cyclin-dependent kinase 2 (CDK2), Cyclin E1 (CCNE1), p21, and cell division cycle 25A (CDC25A) [42,43]. Furthermore, p53 is crucial in controlling the metabolic reprogramming of tumors. Research has shown that the transcription factor p53 can regulate nicotinamide adenine dinucleotide phosphate (NADPH) production and glutamine metabolism in tumor cells by inhibiting the expression of malic enzyme 1 (ME1) and malic enzyme 2 (ME2) related to the tricarboxylic acid cycle (TCA cycle), thereby influencing the growth of tumor cells. In contrast, the downregulation of ME1 and ME2 reciprocally activates p53 in a feedforward manner through distinct mechanisms mediated by murine double minute 2 (Mdm2) and adenosine monophosphate-activated protein kinase (AMPK) [44]. Similarly, p53 can modulate ammonia metabolism in tumor cells by restraining the urea cycle, thereby impeding urea production and ammonia elimination both in vitro and in vivo, ultimately inhibiting tumor growth. Similarly, downregulation of the urea cycle metabolic enzyme genes can activate p53 through Mdm2-mediated mechanisms [45]. In addition, the tumor suppressor protein p53 directly interacts with glucose 6-phosphate dehydrogenase (G6PD), a pivotal enzyme in the pentose phosphate pathway (PPP), hindering the active dimer formation of G6PD and, consequently, restraining the growth of tumor cells [46]. Unfortunately, despite the capability of wild-type p53 to impede tumor cell growth and carcinogenic transformation, it is susceptible to inactivation and mutation in human tumors [47]. p53 mutations not only compromise its anti-tumor activity but also confer carcinogenic properties on the mutated p53 protein. This dual impact involves a loss-of-function (LOF) required for tumor inhibition and a gain-of-function (GOF) required for tumor growth, thus promoting tumor proliferation and metastasis [48]. Therefore, *TP53*, an essential yet highly mutated tumor suppressor gene, stands out as an appealing therapeutic target for cancer therapy.

## 3. Formation and Factors of the Tumor Microenvironment

Earlier studies found that mutations in oncogenes or tumor suppressor genes activate tumor signaling pathways, resulting in abnormal cell proliferation and migration processes. However, it was gradually realized that tumors are more than just a hereditary condition; they are the result of multiple interactions between tumor cells and non-cancer cells in a very complex environment [49]. The tumor microenvironment is a highly structured ecosystem that includes tumor cells, immune cells, cancer-related fibroblasts (CAFs), endothelial cells (ECs), parietal cells, and other tissue-resident cells embedded in the altered and angiogenic ECM [49]. The contacts and signal crosstalk between cancer cells and immune cells eventually create an environment that encourages tumor development and spread. TME consists of cellular components (neutrophils, eosinophils, basophils, T cells, B cells, macrophages, myelogenic suppressor cells, natural killer cells, cancer stem cells, fibroblasts, mesenchymal stem cells, adipocytes, nerves, vascular endothelial cells, and pericytes) and non-cellular components (chemokine, interleukin, growth factor, oxygen, extracellular matrix protein, pH value, etc.) [50,51]. To sustain proliferation and adapt to adverse conditions, tumor cells actively alter their microenvironment by secreting various cytokines, chemokines, and other factors [7]. This not only results in alterations to their intrinsic signaling pathways but also promotes cellular reprogramming. Apart from direct cell-to-cell contact, non-cellular elements within the tumor microenvironment may facilitate intercellular communication via many signaling pathways when tumor cells are under stress or experience aberrant signal activation. These elements bind to receptors or alter the ECM to have both direct and indirect effects on target cells [52]. These intricate cellular interactions contribute to the establishment and dynamic evolution of the tumor microenvironment.

Significantly, alterations in metabolic requirements and the secretion of metabolites by cellular components within both tumor cells and the tumor microenvironment contribute to the establishment of a conducive environment for tumor growth. This has the capacity to adapt to a distinct metabolic phenotype, profoundly impacting the TME [8]. Through metabolic reprogramming, tumor cells release substantial quantities of metabolites into the TME while consuming energy substances, thereby amplifying the complexity of non-cellular components in the TME [53]. Moreover, the proliferation of tumors and immune cells results in a competition for essential nutrients crucial for anti-tumor defense, leading to nutrient deficiency within the TME. This challenging milieu compels infiltrating immune cells to undergo metabolic adaptations associated with tolerance phenotypes [54]. Consequently, these metabolic changes in immune cells critically compromise the efficacy of the anti-tumor immune response. In addition to changes in metabolic pathways caused by abnormal proliferation of tumor cells, immune cells also develop specific metabolic characteristics during activation, adaptation to different tissue environments, and inflammation or disease [55]. The disruption of these metabolic pathways has the potential to reshape the destiny of immune cells, influencing immune regulation. Damaged immune cells can accelerate immune escape and the migration of tumor cells. In general, immune cells within the tumor microenvironment and tumor cells engage in both immunological monitoring of the tumor cells and crosstalk of signaling molecules (Figure 2). In summary, the intrinsic characteristics of tumor cells, including genetic alterations, epigenetic alterations, metabolic reprogramming, and signal release, influence how tumors mold their microenvironment. The transformation of normal cells into malignant cells entails overcoming various hurdles to initiate tumor formation. These hurdles include evading immune cell attacks within the TME and reshaping the surrounding matrix into a TME that fosters tumor support, ensuring the fulfillment of oxygen and nutrient supply requirements.

## 4. Signaling in the Tumor Microenvironment

The tumor microenvironment can activate important signaling molecular pathways in tumor cells through direct cell–cell contact or the secretion of soluble factors that are essential for cell survival. The interplay between dysregulated signals from tumor cells and the environment encompasses numerous pathways involving both autocrine and paracrine signaling [56]. Among them, abnormal proliferation and metabolic reprogramming of tumor cells induce alterations not only in intercellular communication but also in the non-cellular components within the TME. These changes influence signal transmission within the TME and impact the growth of immune cells. In addition, non-cellular components in the TME, such as growth factors, oxygen, lactic acid, etc., can impede the growth and signal transmission of immune cells (Figure 3). This interference contributes to the immune escape process of tumor cells through a feedback mechanism [6]. Therefore, signal transduction within the tumor microenvironment not only directly influences the growth of tumor cells but also mediates the immune escape process of tumor cells by interfering with the function of immune cells.

### 4.1. Metabolite-Mediated Signaling

Lactic acid is a metabolite of tumor cells and serves as a signaling molecule. The exchange of lactate within the TME and tumor cells is mediated by the monocarboxylate transporter-1 (MCT1) and monocarboxylate transporter-4 (MCT4). Among them, MCT1 is responsible for the uptake of exogenous lactic acid, while MCT4 mainly outputs lactic acid produced during glycolysis [57,58]. The excessive accumulation of lactic acid in the TME inhibits the activity of the protein responsible for hydroxylating hypoxia-inducible factor-1α (HIF-1α) prolyl hydroxylase domain (PHD), indirectly leading to the ubiquitin-mediated degradation of HIF-1α [59]. Angiogenesis promotes tumor progression and is primarily driven by hypoxia and the hypoxia-inducible factor 1 (HIF-1) pathway. Lactic acid plays a role in stabilizing HIF-1α, thereby mediating the promotion of angiogenesis and tumor growth. Additionally, lactic acid can independently induce hypoxia without involving HIF-1. It directly binds to various oxygen regulatory proteins, such as the N-Myc downstream-regulated gene (NDRG) family member N-Myc downstream regulatory gene 3 (NDRG3), preventing its degradation [60]. The accumulation of NDRG3, in turn, induces angiogenesis by activating ERK signal cascades. The excessive accumulation of lactic acid, a protein post-translational modification donor, within both tumor cells and the tumor microenvironment influences the regulation of signaling pathways in tumors. Recent studies indicate that tumor cells metabolize lactic acid, and this metabolic process can facilitate lactate modification, subsequently modulating the activity of meiotic recombination 11 homolog 1 (MRE11). MRE11, a nuclease pivotal in initiating DNA end cleavage and homologous recombination repair, is thereby regulated by the lactic acid-mediated modification, contributing to the homologous recombination repair process in tumor cells [61]. This suggests that metabolites originating from tumor cells, specifically lactic acid, function as a critical molecular link connecting cell metabolism to homologous recombination repair mechanisms. Lactic acid, aside from its regulatory impact on tumor cells, exhibits the ability to trigger apoptosis in natural killer (NK) and natural killer T (NKT) cells [62]. This induction of apoptosis leads to the immune escape of tumor cells. Similarly, lactic acid has also been reported to inhibit mTOR signaling, blocking the production of interferon-γ (IFN-γ) and interleukin-4 (IL-4) by NKT cells within the TME. This inhibition prevents the activation of these immune cells [63]. Other waste products of cell metabolism (kynurenine and adenosine) can also play an immunomodulatory role within the TME. Studies indicate that the metabolites generated following an increase in kynurenine level in the TME induce immunosuppression of T cells [64]. Furthermore, the accumulation of adenosine, the breakdown product of nucleotide metabolism in the TME, effectively reduces the cytotoxic effects mediated by T cells and NK cells, while simultaneously enhancing the activation of the immunomodulatory alternatively activated macrophage (M2) [65].

Glutamine serves as a crucial metabolic substrate that facilitates the rapid proliferation of tumor cells, and a majority of these cells exhibit a pronounced “glutamine addiction”. Carcinogenic changes in tumor cells reprogram glutamine metabolism. For instance, proto-oncogene cellular MYC (c-MYC) transcription binds to promoter regions of high-affinity glutamine input proteins, including alanine-serine-cysteine transporter 2 (ASCT2, sodium-dependent neutral amino acid transporter type 2, also known as solute carrier transporter 1A5 (SLC1A5)) and skip N2 metastases (SN2) (Subtype of system N, also known as selective amino acid transporters in micropinocytosis (SLC38A5)). This results in tumor cells absorbing more glutamine [66]. Similarly, carcinogenic KRAS enhances gene expression of enzymes related to glutamine metabolism in KRAS-transformed cells, thus promoting tumor cell proliferation [67]. On the contrary, the tumor suppressor p53 plays a role in activating the expression of glutaminase 2 (GLS2) while concurrently eliminating intracellular reactive oxygen species (ROS). This dual action serves to safeguard cells from genomic damage, thereby thwarting the initiation of cancerous transformations [68]. In addition, the changes in glutamine metabolism in tumor cells, various cells, and cytokines in the TME disrupt glutamine metabolism through diverse signaling pathways. Immune cells’ secretion of interleukin-4 (IL-4) is known to enhance the expression level of the glutamine transporter ASCT2 in breast cancer cells [69]. Similarly, IL-4 acts to promote glutamine uptake through the Janus kinase (JAK)/STAT pathway. These findings imply that the interplay of glutamine metabolism between tumor cells and the TME significantly impacts tumor cell growth.

Adipocytes within the TME metabolize fatty acids, offering a crucial nutrient source for cancer cell proliferation [70]. Since fat is almost ubiquitous throughout the body, adipocytes emerge as prevalent components within the TME of most solid tumors. In addition to providing energy for tumor cells, fatty acids function as precursors for the structural constituents of newly synthesized membranes and lipid signaling molecules, such as phosphatidylinositol-3,4,5-trisphosphate (PIP3) and lysophosphatidic acid (LPA). Furthermore, fatty acids activate AKT, thereby influencing the regulation of tumor cell metabolism and migration [71]. In parallel, akin to other metabolites present in the TME, fatty acids play a role in modulating immune processes within tumors. For instance, saturated fatty acids can stimulate the production of proinflammatory cytokines (for example, interleukin-23 (IL-23) during the IL-17-producing T helper (Th17) response). Conversely, polyunsaturated fatty acids are associated with the induction of anti-inflammatory cytokines (for example, interleukin-10 (IL-10)) [72]. In addition, fatty acids play a coordinating role in triggering ferroptosis in tumor cells and modulating the immune response involving CD8^+^ T cells through acyl-CoA synthetase long-chain family member 4 (ACSL4) [73].

A tumor-derived exosome (TDE) either initiates or inhibits various signaling pathways in recipient cells by delivering heterogeneous cargo, thereby participating in a series of biological processes such as TME remodeling, angiogenesis, invasion, metastasis, and drug resistance [74]. The contents of exosomes consist of various proteins found inside and on their surface, encompassing receptors, transcription factors, enzymes, extracellular matrix proteins, lipids, and nucleic acids (DNA, mRNA, and miRNA), which constitute their contents [75]. Exosomes carry nucleic acids and proteins that can alter the destiny of recipient cells via autocrine and paracrine signaling. First, exosomal proteins can influence the fate of the cell that releases the exosome cell itself through an autocrine pathway. For example, exosomes released by chronic myeloid leukemia cells contain a cytokine transforming growth factor-β1 (TGFβ1), which binds to TGFβ1 receptors on leukemia cells. This binding activates ERK, AKT, and anti-apoptotic pathways to promote tumor cell growth [76]. In comparison to autocrine effects, exosome-mediated paracrine mechanisms involving cell-to-cell interaction and microenvironmental regulation are also implicated in the regulation of tumor growth. In glioma cells, the transmission of extracellular vesicles (EVs) containing the oncogenic receptor epidermal growth factor receptor variant III (EGFRvIII) to adjacent glioma cells lacking the receptor triggers the activation of the AKT pathway. This activation imparts to adjacent glioma cells the ability to invade and metastasize [77]. Similarly, the transfer of the mutated *KRAS* gene, along with other oncogenes such as *EGFR*, from colon cancer cells to recipient cells with wild-type KRAS occurs through exosomes, promoting tumor invasion [78]. Taken together, metabolites from tumor cells and immune cells within the TME can regulate tumor cell growth and immune escape processes through various signaling pathways.

### 4.2. Signal Transduction Mediated by Soluble Factors

It is well known that cytokines and growth factors produced by stromal cells have a dramatic impact on tumor cells. Diverse cells within the tumor microenvironment are capable of secreting various cytokines and growth factors, including stromal cell-derived factor-1 (SDF-1), interleukin-6 (IL-6), vascular endothelial-derived growth factor (VEGF), insulin-like growth factor 1 (IGF-1) and hepatocyte growth factor (HGF) [79]. These receptor signals facilitate the growth, metastasis, and survival of tumor cells by interacting with ligands that are abundantly expressed in tumor cells. Certain types of solid tumor cells use stromal cell-derived factor 1α (SDF-1α) in conjunction with their C-X-C chemokine receptor type 4 (CXCR4) to cause cancerous cells to spread to common locations including the lungs, lymph nodes, and bone marrow [80]. Neovascularization plays a crucial role in tumor development. Upon reaching a size of 1–2 mm, a tumor necessitates the establishment of a vascular system to facilitate the delivery of oxygen and nutrients [81]. Hypoxia, characterized by insufficient oxygen levels in tissues, serves as a primary stimulus for angiogenesis. Many molecules responsive to hypoxia play a role in promoting angiogenic switching, with VEGF and its downstream signaling pathways being predominant drivers [5]. Members of the VEGF family specifically induce endothelial cell division, proliferation, and migration, exhibiting dual functions in promoting both angiogenesis and increased vascular permeability. Tumor cells secrete VEGF, which interacts with VEGF receptors on neighboring stromal vascular endothelial cells. This interaction stimulates the division and proliferation of vascular endothelial cells, initiating tumor angiogenesis. Consequently, it enhances vascular permeability, facilitating tumor growth and metastasis—a phenomenon referred to as the paracrine effect of VEGF [82]. Inhibition of VEGF signaling can prevent angiogenesis and tumor growth in mice [83], underscoring the pivotal role of angiogenesis as a key step in tumorigenesis. Similarly, the IGF-1 axis is a complex molecular regulatory network critical for cell proliferation, growth, and death regulation [84]. Ligands within the insulin-like growth factor (IGF) axis predominantly comprise IGF-1, insulin-like growth factor 2 (IGF-2), and insulin. Notably, IGF-1 primarily binds to the insulin-like growth factor receptor (IGF-1R) and activates the PI3K/AKT and MEK/ERK signaling pathways. Consequently, these pathways modulate the migration, invasion, and distant organ metastasis of tumor cells [85]. Research has identified the widespread expression of IGF-1R in various cell types within the TME, including epithelial cancer cells, CAFs, and myeloid cells, among others [86]. In fact, extensive investigations have delved into the molecular interactions between tumor cells and the microenvironment mediated by soluble factors. For example, in multiple myeloma (MM) cells, IL-6, IGF-1, SDF-1α, tumor necrosis factor-α (TNF-α), and VEGF can promote tumor cell proliferation through the MEK/p42/p44/MAPK signaling cascade [87,88]. In chronic lymphocytic leukemia (CLL), interferon-α (INF-α) and basic fibroblast growth factor (bFGF) prevent tumor cell apoptosis through a Bcl-2-dependent pathway [89,90]. However, IL-4 and interferon-γ (INF-γ) can upregulate the expression of inducible nitric oxide synthase (iNOS) in CLL cells. This results in the endogenous release of nitric oxide (NO), which inhibits tumor cell apoptosis through the S-nitrosation of caspase-3 [91].

### 4.3. Cellular Contact-Mediated Molecular Signaling

Within the TME, various forms of cell-to-cell contact and communication occur during the interaction between cancer cells and stromal cells. These interactions encompass connections facilitated by molecules of cell adhesion devices, engagements between membrane-bound ligands and receptors, and contacts mediated by tunneling nanotube (TNT) and tumor microtube (TM) [92]. Through these interactions, invasive cancer cells typically establish contact with stromal cells through cell adhesion molecules at adhesion junctions [93]. The adhesion junctions and gap junctions play pivotal roles in facilitating direct connections between tumor cells and stromal cells. Adherent junctions primarily consist of transmembrane cell adhesion molecules such as cadherins and adherins, along with scaffold proteins [94]. During the epithelial–mesenchymal transition, a shift in E-cadherin expression to N-cadherin expression in tumor cells intensifies fibroblast growth factor signaling, thereby promoting tumor cell invasion and metastasis. Moreover, this signal transduction also increases the expression and secretion of matrix metalloproteinase-9 along with enhancing physical contact between cancer cells and the endothelium and matrix [95,96]. Additionally, within the TME, direct contact between tumor cells and stromal cells is facilitated through interactions between membrane-tethered ligands and their respective receptors. A notable example involves the interaction between programmed cell death ligand-1 (PD-L1) in tumor cells with its receptor programmed cell death protein 1 (PD-1) in effector T cells. This interaction inhibits T cell-mediated anti-tumor immune responses, thereby promoting the growth of tumor cells [97]. Moreover, this interaction has the potential to upregulate the expression of P-glycoprotein through the mitogen-activated protein kinase pathway downstream of PI3K/AKT and PD-L1, consequently conferring drug resistance to tumor cells [98]. Notably, TNT and TM represent innovative modes of information communication between tumor cells and normal cells within the TME. These structures enable long-distance direct communication between cancer cells and stromal cells. TNT and TM are filamentous channels composed of F-actin, facilitating rapid exchange between tumor cells and other cellular components and molecules, including organelles, vesicles, molecules, and ions [99]. This suggests that the molecules mediating the interaction between cancer cells and stromal cells will exert either tumor-suppressive or tumor-promoting functions in the tumor microenvironment.

### 4.4. Transduction of ER Stress Signals

The endoplasmic reticulum (ER) serves as a crucial site for processing, modifying, and folding proteins, playing a pivotal role in the tightly regulated processes that govern cell function, fate, and survival. The accumulation of misfolded proteins in the ER lumen causes ER stress, which activates unfolded protein response (UPR). The UPR involves three signaling pathways: inositol-requiring enzyme-1α (IRE1α), protein kinase RNA-like ER kinase (PERK), and activating transcription factor 6 isoform α (ATF6α) [100]. The activation of oncogenic signaling pathways and abnormal metabolic changes in tumor cells will lead to changes in various components in the TME, including hypoxia, nutrient deprivation, and the accumulation of acid and peroxides, creating harsh conditions. These changes further disrupt endoplasmic reticulum (ER) homeostasis in both malignant and stromal cells, leading to sustained ER stress and unfolded protein response (UPR). Studies have shown that glucose limitation in the TME impacts adenosine 5’-triphosphate (ATP) production in tumor cells. The reduced ATP levels affect the energy sources and phosphate donors necessary for protein folding in the ER [101]. Similarly, insufficient availability of amino acids represents another significant stressor in TME. Amino acid starvation triggers the activation of the general control nonderepressible 2 (GCN2) kinase. This activation leads to the phosphorylation of eukaryotic initiation factor 2 subunit α (eIF2α) and activates the integrated stress response (ISR). This response allows tumor cells to adapt to stress within the TME [102]. In addition, the accumulation of reactive oxygen species (ROS) in the TME results from various signal transduction events. Excessive intracellular ROS accumulation can cause protein folding stress by changing the calcium channels within the endoplasmic reticulum where it resides [103]. Moreover, the stress response of tumor cells to the endoplasmic reticulum can affect the malignant progression of tumors by changing the function of immune cells coexisting in TME. For instance, the induction of endoplasmic reticulum stress and activation of the unfolded protein response (UPR) can impede the surface expression of major histocompatibility complex class I (MHC-I) molecules. This effect is achieved through the upregulation of X-box binding protein 1 (XBP1) and activating transcription factor 6 (ATF6) [104]. Breast cancer cells exhibiting over-activation of IRE1α can stimulate the production of proinflammatory and immunomodulatory cytokines, including IL-6, interleukin-8 (IL-8), C-X-C motif chemokine ligand 1 (CXCL1), and granulocyte-macrophage colony-stimulating factor (GM-CSF) [105]. In contrast, the ablation of IRE1α reduces the accumulation of CAFs and myeloid-derived suppressor cells (MDSC), while restructuring the TME in triple-negative breast cancer (TNBC) [106]. These studies imply that the abnormal activation of ER stress sensors and their downstream signaling pathways has become a pivotal regulator of tumor growth and metastasis, and response to chemotherapy, targeted therapy, and immunotherapy.

### 4.5. Transduction of Notch Signaling

Numerous pathways contribute to the cross-talk and paracrine signaling between cancer cells and the TME. Notably, the Notch signaling pathway is recognized as a key player in each component of both cancer cells and the TME [56]. The interaction between Notch and its ligands can initiate a signaling cascade that is capable of regulating cell proliferation and apoptosis [107]. The principal components of the Notch signaling system encompass the Notch receptor, Delta-Serrate-Lag2 (DSL) ligands, CCAAT-binding factor (CBF)-1, suppressor of hairless, and lag-2 (CSL) DNA-binding protein. In mammals, DSL ligands include three classes of delta-like ligands (delta-like 1 (DLL1), delta-like 3 (DLL3), and delta-like 4 (DLL4)) and two serrated ligands (Jagged1 (JAG1) and Jagged2 (JAG2)). The Notch intracellular domain (NICD) of the receptor translocates into the nucleus in response to activation, starting transcription activities for target genes downstream. The degradation of NICD in a proteasome-dependent manner may control the termination of Notch signaling [108]. In addition to the crosstalk of signaling pathways expressed by Notch ligands and receptors that would mediate Notch signaling between different compartments of the TME, the IL-6/signal transducer and activator of transcription 3 (STAT3) pathway in the TME would also regulate Notch signaling through interactions between different compartments. For instance, in primary hepatocellular carcinoma, there is an interaction between cancer cells and CAFs via IL-6, STAT3, and Notch [109]. Notch signaling in multiple myeloma also controls the release of IL-6 [110]. Upregulation of JAG1 expression in prostate cancer cells leads to the reactivation of the stroma. This indicates that communication of information between the TME and Notch can occur through the transforming growth factor-β (TGF-β) pathway [111]. Similarly, JAG1/Notch signaling has been shown to regulate many aspects of tumorigenesis, including stem cell development, the epithelial–mesenchymal transition, and immune cell homeostasis during minimal residual disease. Treatment with an anti-JAG1 antibody can inhibit the JAG1/Notch signaling pathway in both tumor cells and the microenvironment, thereby delaying tumor recurrence [112]. Interestingly, exosomes within the TME also impact tumor growth by activating Notch signaling. Recent studies have found that a hardened extracellular matrix can promote the release of tumor exosomes, and Jagged1, abundant in these exosomes, can mediate the activation of the Notch signaling pathway, thereby promoting tumor growth [113]. This suggests that the transmission and activation of Notch signaling within the TME play an important role in regulating tumor cell growth and migration.

## 5. Effects of Signaling within the TME on Immune Cells

TME has garnered growing attention owing to its pivotal involvement in tumor immunosuppression, distant metastasis, local drug resistance, and response to targeted therapy [114]. TME is a place where the bioenergy needs of immune cells and fast-growing tumors struggle for the nutrients required for anti-tumor defense. The reduction of nutrients such as glucose, glutamine, and amino acids will affect the function of immune cells through a series of signaling pathways. However, the challenging conditions within the TME will also force the infiltrating immune cells to undergo metabolic adaptations associated with the tolerance phenotype. Ultimately, these metabolic changes in immune cells can undermine the effectiveness of the anti-tumor immune response. Moreover, metabolic interactions among tumor cells, immune cells, and stromal cells can result in the secretion of certain metabolites or cytokines, leading to changes in the TME. The modified TME, in turn, can influence the function of immune cells and mediate the growth of tumor cells through signaling pathways.

### 5.1. T Cells

The TME has a complex immune cell milieu that includes both innate immune cells (NK cells, macrophages, and dendritic cells) and adaptive immune cells (CD8^+^ and CD4+ T cells). Cytotoxic lymphocytes include NK cells and CD8+ T cells, whereas CD4+ T cells develop into subsets such as Th1, Th2, Th17, and Regulatory T cells (Treg). After recognizing tumor-associated antigens, cytotoxic CD8+ T cells provide a natural defense against tumor progression by specifically killing tumor cells. Studies have found that the glucose concentration in the TME can affect the function of immune cells through various signaling pathways. For example, an excess of glucose promotes increased production of reactive oxygen species in the T cells’ mitochondria. This activation triggers certain TGF-β Retinoic acid receptor-related orphan receptor γ (RORγ) T cytokines related to Th17, leading to excessive differentiation of Th17 cells and triggering inflammation in vivo [115]. However, in the absence of glucose within the tumor microenvironment (TME), the functionality of most immune cells becomes defective. Low glucose levels activate the “energy receptor” AMPK within cells, leading to the phosphorylation of the mTORC1 receptor and subsequent inhibition of the mTORC1 signaling pathway [116]. In addition to glucose, the accumulation of fumarate in the TME inhibits zeta-chain-associated protein kinase (ZAP70), a crucial kinase in the T-cell receptor (TCR) signaling pathway. This inhibition affects the activities of ZAP70 and TCR signaling pathways, impeding the activation of CD8+ T cells and anti-tumor immune function [117]. Similarly, TCR-dependent CD8+ T cell activation is linked to the stimulation of one-carbon metabolism. Low levels of serine in the TME result in CD8+ T cell anti-tumor dysfunction. However, supplementation of one carbon unit of formate within the TME can enhance the function of exhausted CD8+ T cells [118]. Recent research has demonstrated that intracellular and extracellular pH (pHi/pHe) play an important role in controlling T cell activity. The TCR-induced inhibitor of the T-cell receptor signaling protein complex (STS1-Cbl-b) senses intracellular or extracellular acidity and regulates T cell responses. A deficiency of TCR signaling 1(STS1) or casitas B lymphoma-b (Cbl-b) can promote T cell proliferation and differentiation activity in vivo, inhibit tumor growth, prolong survival, and improve T cell fitness in tumor models [119].

### 5.2. Macrophages

Macrophages present in the TME are called tumor-associated macrophages (TAMs) and constitute the primary infiltrating immune cells in the TME [120,121]. In a variety of solid tumor types, the abundance of TAMs is associated with poor prognosis, with numerous studies highlighting their immunosuppressive role in tumor progression and spread [122]. TAMs exhibit distinct phenotypes, with the classically activated macrophage (M1) having anti-tumor functions and M2 having pro-tumor functions. Research indicates that other soluble factors, such as cytokines, and chemokines, secreted by neighboring cells within the TME influence the polarization of TAMs to M1 or M2 phenotypes. For example, NR_109, functioning as a long non-coding RNA, can establish a signaling pathway (NR_109/FUBP1/c-Myc axis) in conjunction with far upstream element binding protein 1 (FUBP1). This pathway regulates the polarization of TAMs, reconstructs the tumor microenvironment, and fosters cancer development [123]. Lactic acid, a metabolite in the TME, can stabilize HIF-1α mediated VEGF expression. Moreover, it can activate G-protein-coupled receptor 132 (GPR132), inducing the differentiation of TAMs into the M2 subtype and facilitating breast cancer metastasis [124]. In TME, there is an upregulation in the expression of N-methyl-D-aspartate receptors (NMDARs) in macrophages. Activation of these receptors induces calcium influx, subsequently activating the downstream calcium-calmodulin (CaM)-dependent protein kinase II (CaMKII)/ERK/cAMP response element-binding protein (CREB) pathway. This activation leads to oxidative phosphorylation, an increase in ROS levels, and disruption of mitochondrial structure in macrophages. Consequently, this results in an enhanced immunosuppressive function of TAMs [125]. Recent studies have found that the high accumulation of citrulline, a urea cycle metabolite, within the TME facilitates the direct interaction between citrulline and Janus kinase 2 (JAK2). This interaction weakens the binding of JAK2 to IFNγ receptor beta chain (IFNγR) and the signal transducer and activator of transcription 1 (STAT1), and it inhibits the activity of the JAK2-STAT1 signaling pathway. Consequently, this inhibition suppresses the proinflammatory polarization of macrophages and the resistance to infection in mice [126]. Interestingly, TAMs can have direct effects on the metabolism of tumor cells. contribute to the increased glycolysis of non-small-cell lung cancer (NSCLC) cells by secreting TNF-α. Simultaneously, TAMs enhance hypoxia in the TME by activating AMP-activated protein kinase and Peroxisome proliferator-activated receptor-gamma co-activator-1α (PGC-1α). In addition, the depletion of TAMs increases T-cell infiltration and PD-L1 expression in tumor cells, which is beneficial for PD-L1 blockade in NSCLC [127]. A positive feedback loop exists between macrophages and the hypoxic environment, and hypoxia will drive TAMs polarization, while TAMs drive the hypoxic environment within the TME through adverse angiogenesis, thereby intervening in the tumor immune process [127].

### 5.3. Stromal Cells

CAFs represent a diverse group of stromal cells associated with tumors. Constituting the most abundant cell type in the TME, CAFs play a crucial role by secreting various cytokines, extracellular vesicles, and other substances that closely influence ECM remodeling, as well as the migration, invasion, and immune evasion of tumor cells [128]. First, various factors in the TME induce CAF activation by stimulating certain different signaling pathways during CAF generation. For example, inflammatory mediators such as interleukin-1β (IL-1β) and IL-6 act through the noncanonical nuclear factor-κB (NF-κB) and JAK/STAT3 pathways, respectively, to promote the malignant progression of CAFs [129]. Significantly, the interplay between CAFs and immune cells, along with other immune constituents, can modulate the TME, consequently dampening anti-tumor immune responses. In the context of colorectal cancer (CRC), the secretion of C-X-C motif chemokine ligand 1 (CXCL5) secreted by CAFs binds to C-X-C chemokine receptor type 2 (CXCR2) on tumor cells, activating the PI3K/AKT signaling pathway. This activation results in the promotion of PD-L1 expression on the surface of tumor cells, ultimately leading to immune evasion [130]. Furthermore, CAF-derived wingless-type MMTV integration site family member 2 (WNT2) inhibits the p-JAK2/p-STAT3 (TYR705) pathway by upregulating the suppressor of cytokine signaling 3 (SOCS3) inhibitors on dendritic cell (DC) precursors. This action effectively obstructs DC differentiation and maturation [131]. Notably, the ECM is a complex network of extracellular proteins, proteoglycans, and glycoproteins. Serving as a pivotal component within the TME, the ECM is frequently implicated in cancer progression [132]. Studies have found that CAFs possess the capability to enhance the degradation of the normal ECM structure. They achieve this by secreting various matrix proteins such as fibronectin and type I collagen and generating an assortment of matrix metalloproteinases, including matrix metalloproteinase-1 (MMP-1) and matrix metalloproteinase-1 (MMP-3) [133,134]. This underscores the indispensable role of CAFs in orchestrating ECM remodeling.

Fat is ubiquitous throughout the body and distributed extensively. Within the TME, stromal adipocytes serve as a reservoir of energy for tumors, actively contributing to tumorigenesis through anabolic processes as well as functioning as endocrine organs. Adipocytes secrete various substances (such as adipokines and bioactive lipids) in the TME which can promote tumor progression, metastasis, and drug resistance. A pivotal adipokine in this context is leptin, released by mature adipocytes. Leptin can bind to receptors on tumor cells and trigger effects on proliferation, migration, and tumor invasion. For example, leptin will not only promote TAMs through NF-κB/nuclear factor-κB1 (NF-κB1) but also the expression and secretion of interleukin-18 (IL-18) in breast cancer cells through the PI3K/AKT/ATF2 pathway. This cascade ultimately results in the migration and invasion of breast cancer cells [135]. Moreover, the accumulation of leptin in the TME can significantly inhibit CD8+ T cell function by activating the STAT3- fatty acid oxidation (FAO) axis and reducing glycolysis [136]. Leptin derived from adipocytes also promotes Plasminogen activator inhibitor-1 (PAI-1)-mediated breast cancer metastasis in a STAT3/miR-34a-dependent manner [137]. In addition to being able to transfer to cancer cells as a source of nutrients for tumor growth, fatty acids can also act as precursors for the structural units of newly synthesized membrane and lipid signaling molecules (such as phosphatidylinositol-3,4,5-triphosphate). They can activate AKT, thereby regulating the metabolic reprogramming of tumor cells, cell proliferation, and survival [71]. In addition, β-hydroxybutyrate (BHB) serves as a matrix-induced signal, subsequently exerting epigenetic control over gene expression. BHB derived from adipocytes increases the acetylation of global histone H3 lysines 9 (H3K9) in breast cancer cells, thereby upregulating tumor-promoting genes and driving tumorigenesis [138].

## 6. Summary and Prospect

Many signaling pathways in tumor cells interconnect to form a complex network. In the face of exogenous stimulation stress or endogenous oncogene mutations, these signaling pathways will be out of control and confer abnormal proliferation, migration, and invasion abilities on tumor cells. Throughout tumor development, to sustain the capacity for rapid proliferation and adapt to a challenging environment, tumor cells frequently activate or disrupt specific signaling pathways through distinct oncogenes. This process frequently involves the metabolic reprogramming of tumor cells. However, the TME is an intricate ecosystem consisting of cellular compartments with stromal fibroblasts, infiltrating immune cells, blood, and lymphatic vascular networks, and non-cellular components (growth factors, chemokines) including the ECM. Metabolites produced by tumor cells not only act as signaling molecules inside the tumor cells but also are secreted into the TME as cytokines, modifying the activity and function of immune cells and indirectly disrupting tumor progression. Therefore, understanding the roles of secreted metabolites beyond their role in energy production and replication requirements is critical for our understanding of signaling crosstalk in the TME. The transmission of information or communication among cellular components in the TME can occur through direct contact or secretion of cytokines, forming the basis for intricate signaling within the TME.

The challenging conditions within the TME also force metabolic adaptations in infiltrating immune cells, and these metabolic alterations can undermine the effectiveness of anti-tumor immune responses. In recent years, metabolism-mediated chromosome remodeling and non-histone modifications have emerged as crucial factors for comprehending genomic instability and the abnormal proliferation of tumor cells. Metabolites originating from both tumor cells and immune cells can influence the growth and functionality of these cells through diverse signaling pathways after accumulation in the TME. Currently, numerous inhibitors or antagonists target the proliferation signaling pathway and metabolic reprogramming pathway within tumors, but the therapeutic effect is not good. This is largely due to the complexity of crosstalk between various components within the TME, where alterations in a single pathway may disrupt TME homeostasis and exacerbate tumor progression. Therefore, understanding the complexity of the cascade of signaling pathways in tumor cells and the TME, as well as finding metabolites or cytokines susceptible to interference to enhance the anti-tumor ability of immune cells, may serve as a breakthrough in cancer treatment.

## Figures and Tables

**Figure 1 biomolecules-14-00438-f001:**
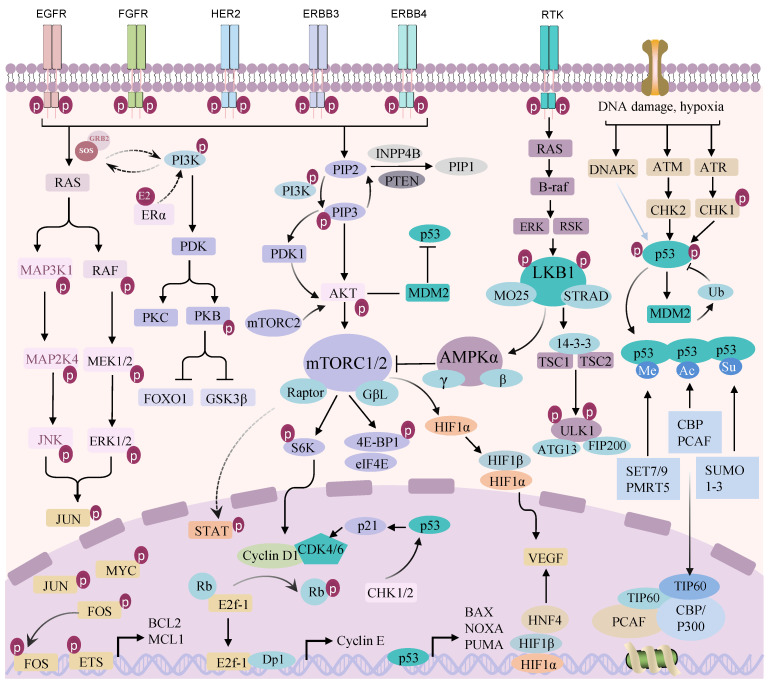
Schematic diagram of the main signaling pathways within cancer cells. The ErbB/EGFR signaling pathway is abnormally activated, which can couple with the PI3K/AKT/mTORC and p53 signaling pathways, thereby regulating the growth and migration of tumor cells. Rb: retinoblastoma protein; E2F1: E2F transcription factor 1; DP1: D prostanoid receptor subtype 1; CHK1: checkpoint kinase 1; CHK2: checkpoint kinase 2; INPP4B: Inositol Polyphosphate 4-phosphatase type II; PTEN: phosphatase and tensin homolog; PIP1: plasma membrane intrinsic protein 1; MDM2: murine double minute 2; AMPKα: AMP-activated protein kinase α; HIF-1α: hypoxia inducible factor-1α; HIF-1β: hypoxia inducible factor-1β; VEGF: vascular endothelial-derived growth factor; HNF4: hepatocyte nuclear factor 4; Bax: BCL2-Associated X; NOXA: phorbol-12-myristate-13-acetate; PUMA: p53 upregulated modulator of apoptosis; B-raf: B-Raf proto-oncogene; ERK: extracellular signal-regulated kinase; RSK: Ribosomal S6 kinase; LKB1: liver kinase B1; MO25: mouse protein-25; STRAD: STE20-related adaptor; TSC1: tuberous sclerosis complex subunit 1; TSC2: tuberous sclerosis complex subunit 2; ULK1: UNC-52-like kinase 1; ATG13: autophagy-related protein 13; FIP200: FAK family kinase-interacting protein of 200 kDa; DNAPK: DNA-dependent protein kinase; ATM: ataxia-telangiectasia mutated proteins; ATR: ataxia telangiectasia mutated and Rad3-related; CHK1: checkpoint kinase 1; CHK2: checkpoint kinase 2; Ub: ubiquitin; Me: methylate; Ac: acetylation; Su: SUMO, small ubiquitin-like modifier; CBP: CREB binding protein; PCAF: p300/CBP-associated factor; SET7/9: (su(var)-3–9,enhancer-of-zeste,trithorax) domain-containing protein 7/9; PRMT5: Protein arginine methyltransferase 5; SUMO1–3: small ubiquitin-like modifier 1–3; TIP60: tat-interacting protein 60; PCAF: P300/CBP-associated factor.

**Figure 2 biomolecules-14-00438-f002:**
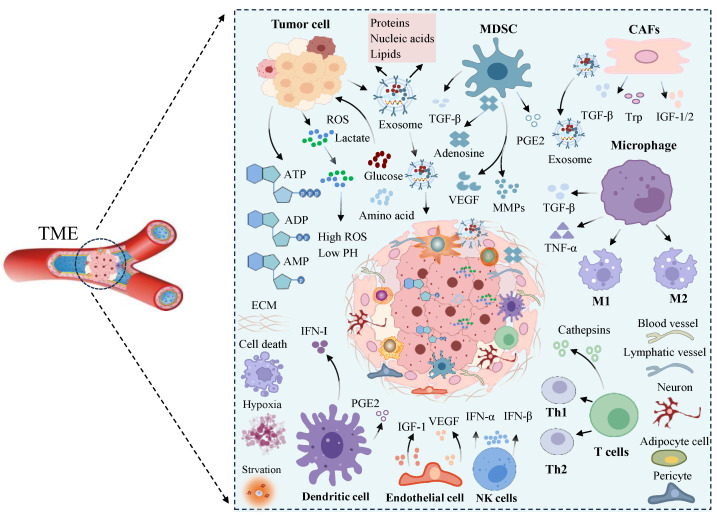
Basic substances for the formation of a complex tumor microenvironment. Blood vessels, tumor cells, immune cells, and various cytokines and metabolites coexist, and the cellular and noncellular components in this special environment interact with each other, forming a complex ecosystem of tumor microenvironment. TME: tumor microenvironment; MDSC: myeloid-derived suppressor cell; CAFs: cancer-related fibroblasts; ROS: removes intracellular reactive oxygen species; ATP: adenosine 5’-triphosphate; ADP: adenosine diphosphate; AMP: adenosine monophosphate; ECM: extracellular matrix; IFN-I: interferon-I; PGE2: Prostaglandin E2; IFN-α: interferon-α; IFN-β: interferon-β; VEGF: vascular endothelial-derived growth factor; TGF-β: transforming growth factor-β; TNF-α: tumor necrosis factor-α; Trp: Tryptophan; IGF-1: insulin like growth factor 1; IGF-2: insulin like growth factor 2; M1: classically activated macrophage; M2: immunomodulatory alternatively activated macrophage; MMPs: matrix metalloproteinases; PGE2: prostaglandin E2.

**Figure 3 biomolecules-14-00438-f003:**
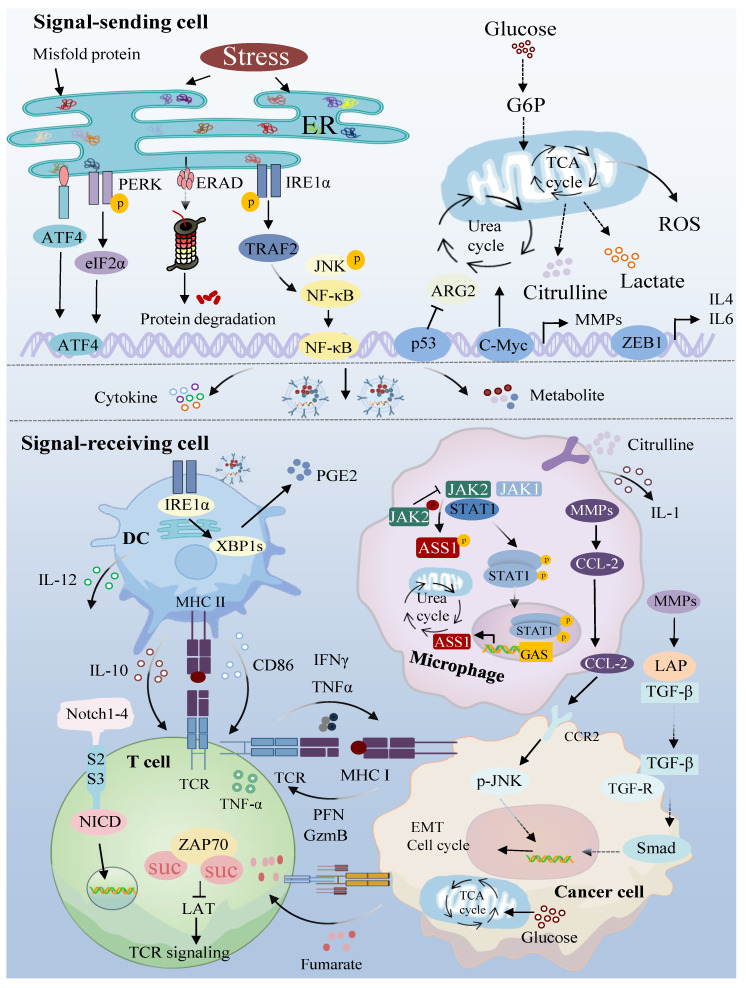
Information exchange and signal transmission between cells within the tumor microenvironment. The metabolic signaling pathways of endoplasmic reticulum stress and dysregulation mediate the production of exosomes and cytokines and regulate the function of immune cells through various signaling pathways in the microenvironment. ER: endoplasmic reticulum; PERK: protein kinase RNA-like ER kinase; ERAD: Endoplasmic reticulum (ER)-associated degradation; IRE1α: inositol-requiring enzyme-1α; ATF4: activating transcription factor 4; eIF2α: eukaryotic initiation factor 2 subunit α; TRAF2: TNF receptor associated factor 2; JNK: c-Jun N-terminal kinase; NF- κb: nuclear factor-κB; PGE2: prostaglandin E2; XBP1s: X-box-binding protein 1; DC: dendritic cell; IL-10: interleukin-10; IL-12: interleukin-12; MHC-I: major histocompatibility complex class I; MHC-II: major histocompatibility complex class II; CD86: Cluster of Differentiation 86; TCR: T-cell receptor; TNF-α: tumor necrosis factor-α; NICD: Notch intracellular domain; ZAP70: zeta-chain-associated protein kinase 70; IFN-γ: interferon-γ; PFN: proximal femoral nails/Perforin/plasma fibronectin; GZMB: granzyme B; G6P: glucose-6-phosphate; TCA cycle: tricarboxylic acid cycle; ROS: reactive oxygen species; ARG2: arginase 2; MMPs: matrix metalloproteinases; C-MYC: cellular MYC; ZEB1: zinc finger E-box binding homeobox 1; JAK1: Janus kinase 1; JAK2: Janus kinase 2; STAT1: signal transducer and activator of transcription 1; ASS1: argininosuccinate synthetase 1; GAS: gamma-activated sequence; CCL-2: chemokine (CC-motif) ligand 2; LAP: latency associated protein; TGF-β: transforming growth factor-β; CCR2: CC chemokine receptor 2; p-JNK: phosphorylated (p)-JNK; EMT cell cycle: Epithelial-to-Mesenchymal Transition cell cycle; Smads: drosophila mothers against decapentaplegic proteins.

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
