# Peer review of "The Tumor Microenvironment: Signal Transduction"

_biomolecules, 2024, doi:10.3390/biom14040438_

Round 1

Reviewer 1 Report

Comments and Suggestions for Authors

I found several mistakes:

Repetitions

Lines 72-73: Various external and internal signals activate and integrate signaling pathways, leading to the execution of diverse cellular functions, including the execution of various cellular functions, such as cell growth

Lines: 204-211; 216-223: Signal transduction in the RAF-MEK-ERK pathway is initiated by binding multiple ligands to RTK, especially growth factor receptors such as EGFR. Gtp-loaded RAS recruits and activates RAF, leading to recruitment of RAF to the plasma membrane, dimerization, and subsequent phosphorylation. After the phosphorylation of RAF, the bispecific protein kinase MEK1/2 is activated, which further leads to the phosphorylation of T202/Y204 sites of ERK1 and T183/Y185 sites of ERK2, thus altering the expression of downstream target genes. Mutations in components of the RAS-RAF-MEK-ERK signaling pathway have been linked to different types of cancer [31].

Initiation of signal transduction in the RAF-MEK-ERK pathway occurs through the binding of multiple ligands to RTK, especially growth factor receptors such as EGFR.Gtp- loaded RAS recruits and activates RAF, facilitating its recruitment to the plasma membrane, dimerization, and subsequent phosphorylation. Following RAF phosphorylation, the bispecific protein kinase MEK1/2 becomes activated, leading to the phosphorylation of ERK1 at T202/Y204 and ERK2 at T183/Y185 sites, ultimately influencing the expression of downstream target genes [34, 35]. Mutations in components of the RAS-RAF-MEK-ERK signaling pathway have been associated with various cancer types.

Line: 456 are also implicated in involved in the regulation of tumor growth

Words or puntuaction to be deleted

Line 136: reliability, availability, and serviceability (RAS) [14].

Line 527: A notable example involves the interaction between programmed cell death ligand-1 (PD-L1) in tumor cells interacts with its receptor programmed cell death protein 1 (PD-1) in effector T cells.

Line 190: state., and further, regulate the growth

Line 629: IL-1-producing T helper (Th1), IL-2-producing T helper (Th2)

Line 183: Therefore, PTEN is often considered to be a tumor suppressor. Nevertheless, instances of PTEN expression deletion in tumor cells are commonly observed

Missing references

Line 141: The PI3K/AKT/mTOR signaling system is linked to other signaling pathways to facilitate tumor growth and metastasis, and it is most commonly dysregulated in a variety of cancer types (refs). These include the phosphorylation of intracellular enzyme activity

Missing words

Line 476: Neovascularization plays a crucial role for tumor development. 

Line 722: Adipocytes secrete various substances (such as adipokines and bioactive lipids) in the TME which can promote tumor progression, metastasis, and drug resistance.

Comments on the Quality of English Language

I found several mistakes:

Repetitions

Lines 72-73: Various external and internal signals activate and integrate signaling pathways, leading to the execution of diverse cellular functions, including the execution of various cellular functions, such as cell growth

Lines: 204-211; 216-223: Signal transduction in the RAF-MEK-ERK pathway is initiated by binding multiple ligands to RTK, especially growth factor receptors such as EGFR. Gtp-loaded RAS recruits and activates RAF, leading to recruitment of RAF to the plasma membrane, dimerization, and subsequent phosphorylation. After the phosphorylation of RAF, the bispecific protein kinase MEK1/2 is activated, which further leads to the phosphorylation of T202/Y204 sites of ERK1 and T183/Y185 sites of ERK2, thus altering the expression of downstream target genes. Mutations in components of the RAS-RAF-MEK-ERK signaling pathway have been linked to different types of cancer [31].

Initiation of signal transduction in the RAF-MEK-ERK pathway occurs through the binding of multiple ligands to RTK, especially growth factor receptors such as EGFR.Gtp- loaded RAS recruits and activates RAF, facilitating its recruitment to the plasma membrane, dimerization, and subsequent phosphorylation. Following RAF phosphorylation, the bispecific protein kinase MEK1/2 becomes activated, leading to the phosphorylation of ERK1 at T202/Y204 and ERK2 at T183/Y185 sites, ultimately influencing the expression of downstream target genes [34, 35]. Mutations in components of the RAS-RAF-MEK-ERK signaling pathway have been associated with various cancer types.

Line: 456 are also implicated in involved in the regulation of tumor growth

Words or puntuaction to be deleted

Line 136: reliability, availability, and serviceability (RAS) [14].

Line 527: A notable example involves the interaction between programmed cell death ligand-1 (PD-L1) in tumor cells interacts with its receptor programmed cell death protein 1 (PD-1) in effector T cells.

Line 190: state., and further, regulate the growth

Line 629: IL-1-producing T helper (Th1), IL-2-producing T helper (Th2)

Line 183: Therefore, PTEN is often considered to be a tumor suppressor. Nevertheless, instances of PTEN expression deletion in tumor cells are commonly observed

Missing references

Line 141: The PI3K/AKT/mTOR signaling system is linked to other signaling pathways to facilitate tumor growth and metastasis, and it is most commonly dysregulated in a variety of cancer types (refs). These include the phosphorylation of intracellular enzyme activity

Missing words

Line 476: Neovascularization plays a crucial role for tumor development. 

Line 722: Adipocytes secrete various substances (such as adipokines and bioactive lipids) in the TME which can promote tumor progression, metastasis, and drug resistance.

Author Response

Reviewer 1:

I found several mistakes:

(1)Repetitions:

Lines 72-73: Various external and internal signals activate and integrate signaling pathways, leading to the execution of diverse cellular functions, including the execution of various cellular functions, such as cell growth.

 Author's Response: Thank you very much for the reviewer's suggestions and assistance. We have made revisions to the manuscript. Thank you.

Lines: 204-211; 216-223: Signal transduction in the RAF-MEK-ERK pathway is initiated by binding multiple ligands to RTK, especially growth factor receptors such as EGFR. Gtp-loaded RAS recruits and activates RAF, leading to recruitment of RAF to the plasma membrane, dimerization, and subsequent phosphorylation. After the phosphorylation of RAF, the bispecific protein kinase MEK1/2 is activated, which further leads to the phosphorylation of T202/Y204 sites of ERK1 and T183/Y185 sites of ERK2, thus altering the expression of downstream target genes. Mutations in components of the RAS-RAF-MEK-ERK signaling pathway have been linked to different types of cancer [31].

Initiation of signal transduction in the RAF-MEK-ERK pathway occurs through the binding of multiple ligands to RTK, especially growth factor receptors such as EGFR.Gtp- loaded RAS recruits and activates RAF, facilitating its recruitment to the plasma membrane, dimerization, and subsequent phosphorylation. Following RAF phosphorylation, the bispecific protein kinase MEK1/2 becomes activated, leading to the phosphorylation of ERK1 at T202/Y204 and ERK2 at T183/Y185 sites, ultimately influencing the expression of downstream target genes [34, 35]. Mutations in components of the RAS-RAF-MEK-ERK signaling pathway have been associated with various cancer types.

Author's Response: Thank you very much for the reviewer's suggestions and assistance. We have removed the duplicate parts and adjusted the content of the manuscript and the position of the references. Thank you.

Line: 456 are also implicated in involved in the regulation of tumor growth

Author's Response: Thank you very much for the reviewer's suggestions and assistance. We have made revisions to the manuscript. Thank you.

(2)Words or puntuaction to be deleted:

Line 136: reliability, availability, and serviceability (RAS) [14].

Author's Response: Thank you to the reviewers for pointing out the errors. We have made the necessary corrections.

Line 527: A notable example involves the interaction between programmed cell death ligand-1 (PD-L1) in tumor cells interacts with its receptor programmed cell death protein 1 (PD-1) in effector T cells.

Author's Response: Thank you to the reviewers for pointing out the errors. We have made the necessary corrections.

Line 190: state., and further, regulate the growth

Author's Response: Thank you to the reviewers for pointing out the errors. We have made the necessary corrections.

Line 629: IL-1-producing T helper (Th1), IL-2-producing T helper (Th2)

Author's Response: Thank you to the reviewers for pointing out the errors. We have made the necessary corrections.

Line 183: Therefore, PTEN is often considered to be a tumor suppressor. Nevertheless, instances of PTEN expression deletion in tumor cells are commonly observed

Author's Response: Thank you to the reviewers for pointing out the errors. We have made the necessary corrections.

(3)Missing references:

Line 141: The PI3K/AKT/mTOR signaling system is linked to other signaling pathways to facilitate tumor growth and metastasis, and it is most commonly dysregulated in a variety of cancer types (refs). These include the phosphorylation of intracellular enzyme activity

 Author's Response: Thank you for the suggestions and assistance from the reviewers. No literature has been cited here, and this statement was summarized by the author himself. To avoid misunderstandings, we have corrected the wording of the manuscript without citing any literature.

Missing words

Line 476: Neovascularization plays a crucial role for tumor development.

Author's Response: Thank you to the reviewers for pointing out the errors. We have made the necessary corrections.

Line 722: Adipocytes secrete various substances (such as adipokines and bioactive lipids) in the TME which can promote tumor progression, metastasis, and drug resistance.

Author's Response: Author's Response: Thank you to the reviewers for pointing out the errors. We have made the necessary corrections.

Reviewer 2 Report

Comments and Suggestions for Authors

This manuscript is a comprehensive and well-written review of oncogenic signaling pathways and a survey of factors in the tumor microenvironment that promote tumor initiation and progression. 

In general, the review is very high level and attempts to cover the “waterfront” of oncology.  Probably due to the broad focus of the review a number of relevant research areas are absent in the background and tumor microenvironment sections.  These include the importance of tissue architecture (and not just proliferation) in restricting or promoting tumor initiation (see reviews of MJ Bissell), and the role of contact inhibition as it relates to tissue architecture and to proliferation.  

The oncogenic signaling section doesn’t fit well with main purpose of the review (TME).  If this remains in the review, the authors are encouraged to link known oncogenic pathways to specific tumors and to discuss which pathways have been targeted for management of specific cancers as well as the success of these strategies. 

In the TME section, the authors neglect the important role of hyaluronan in the resistance to and/or progression of a number of tumors (see Gorbonova for review).  As well, inclusion of attempts to modify the ECM either through targeting specific host cells or extracellular matrix receptors/components would strengthen the review and increase readership interest.  Shortening and/or focusing the signaling section would permit a more in-depth assessment of the TME and its potential as a target for managing cancer.

Author Response

Reviewer 2:

Comments and Suggestions for Authors

This manuscript is a comprehensive and well-written review of oncogenic signaling pathways and a survey of factors in the tumor microenvironment that promote tumor initiation and progression.

Author's Response: Thank you for the reviewer's suggestions and assistance. In fact, the purpose of this manuscript is to extensively introduce the signal transduction between various types of cells in the tumor microenvironment, as well as the information exchange between immune cells and tumor cells, with a focus on the signal pathway transduction in the tumor microenvironment. It is undeniable that organizational structure (rather than just proliferation and restriction) does play an important role in limiting or promoting tumor development. However, the purpose of this article is to introduce information exchange between cancer cells and the tumor microenvironment. Therefore, we apologize that we cannot provide an introduction to all biological events involved in tumor development. Thank you.

The oncogenic signaling section doesn’t fit well with main purpose of the review (TME).  If this remains in the review, the authors are encouraged to link known oncogenic pathways to specific tumors and to discuss which pathways have been targeted for management of specific cancers as well as the success of these strategies.

Author's Response: Thank you for the reviewer's suggestions and assistance. The occurrence and development of tumors are closely related to the tumor microenvironment, but the basis for information exchange and signal transduction between tumors and the tumor microenvironment lies in the activation or dysregulation of oncogenic signals. In a sense, it is precisely because of the generation of carcinogenic signals that more communication (including substances and signals) occurs between tumor cells and the microenvironment. Therefore, in the first part of the manuscript, we provided a detailed introduction to the activation of oncogenic signaling pathways and related molecular mechanisms. We further discussed that the communication pathways between cells are the goal of cancer cell management, and in the later part of the manuscript, we introduced the impact of these information exchanges on the occurrence and development of cancer.

In the TME section, the authors neglect the important role of hyaluronan in the resistance to and/or progression of a number of tumors (see Gorbonova for review).  As well, inclusion of attempts to modify the ECM either through targeting specific host cells or extracellular matrix receptors/components would strengthen the review and increase readership interest.  Shortening and/or focusing the signaling section would permit a more in-depth assessment of the TME and its potential as a target for managing cancer.

Author's Response: Thank you for the reviewer's suggestions and assistance. Thank you very much for the comments and assistance of the reviewer. During the writing process of the manuscript, we also found that hyaluronic acid does play an important role in the resistance and progression of many tumors. However, since the goal of this manuscript is to emphasize the information exchange between tumor cells and metabolites and immune substances within the tumor microenvironment through a series of signaling pathways, we did not have much discussion on the modification of ECM by extracellular matrix components (although there were also some involved). It is worth noting that tumor oncogenic signals are the basis for cancer cells to communicate information, so this review provides a detailed introduction to the signal section. Thank you.
